

# Tropical and mid-latitude teleconnections interacting with the Indian summer monsoon rainfall: A Theory-Guided Causal Effect Network approach

Giorgia Di Capua[1,2], Marlene Kretschmer[1], Reik V. Donner[1,3], Bart van den Hurk[2,4], Ramesh Vellore[5], Raghavan Krishnan[5], and Dim Coumou[1,2]

[1]Potsdam Institute for Climate Impact Research, Potsdam, Germany
[2]VU University of Amsterdam, Institute for Environmental Studies, Amsterdam, Netherlands
[3]Magdeburg-Stendal University of Applied Sciences, Magdeburg, Germany
[4]Deltares, Delft, Netherlands
[5]Indian Institute for Tropical Meteorology, Pune, India



*Correspondence to*: Giorgia Di Capua (dicapua@pik-potsdam.de)


**Abstract.**

The Indian Summer Monsoon (ISM) is characterized by alternating active (wet) and break (dry) phases operating at sub-seasonal timescales, and various studies advocate tropical and mid-latitude teleconnection drivers influence the sub-seasonal ISM rainfall variability. One such driver is the circumglobal teleconnection pattern, which is commonly observed during boreal

summer regulating the variability across the mid-latitudes at sub-seasonal time scales. In this study, a two-way interaction between ISM and circumglobal teleconnection is hypothesized and causal discovery algorithms are employed to examine and quantify the interaction linkage. Our analysis shows that there is a robust causal link from the circumglobal teleconnection pattern and the North Atlantic Oscillation (NAO) to ISM rainfall, and also a reverse causal link from the ISM rainfall to the circumglobal teleconnection pattern. Further, by including regional drivers in the framework, we identify the causal links that

represent the internal dynamics associated with the ISM convective activity operating on weekly timescales, e.g., on weekly time scales, there is precedence of enhanced ascent to increased rainfall over the monsoon trough region which is followed by strong stabilization and convective inhibition. In our analyses, this internal ISM dynamics has the strongest effect, which is about twice as large as those of the mid-latitudes and of tropical MJO variability on the ISM dynamics. With our theory-guided causal effect network approach, we can (1) test physical hypotheses, (2) exploratively search for causal links and (3) quantify

their relative causal contributions. This paves the way for improved (sub)seasonal forecasts.

## 1 Introduction

The Indian summer monsoon (ISM) is crucial for the Indian society, which receives 75% of its total annual rainfall from the summer months June through September (JJAS). The ISM rainfall variability at sub-seasonal time-scales is characterized by



periods of enhanced and reduced rainfall activity over the monsoon-core region of central India. These periods are usually referred as active (wet) and break (dry) phases respectively. Prolonged active and break spells in the ISM can lead to floods or droughts and consequently have severe socio-economic implications for the Indian subcontinent. A salient semi-permanent

feature of the ISM is the "monsoon trough" (MT) which manifests as a low pressure zone extending from northwestern India into the Gangetic plains and the Bay of Bengal (Rao, 1976; Krishnamurti and Sugi, 1987; Choudhury and Krishnan, 2011). The rainfall amount over the MT region is generally used to define dry and wet spells within the ISM season (Krishnan et al., 2000; Gadgil and Joseph, 2003). The position and strength of MT significantly influences the spatial distribution of monsoon precipitation and associated agricultural productivity on the Indian subcontinent, and the internal dynamics of the ISM

circulation itself provides a first mechanism for sub-seasonal rainfall variability (Pathak et al., 2017).

While the land-sea temperature difference and the mid-tropospheric thermal forcing over the Tibetan Plateau are the prime drivers for the monsoon circulation (Yanai and Wu, 2006), ascending motions over the Indian subcontinent enhance the northward air flux from the ocean toward the land thereby bringing moisture from the ocean and fuelling rainfall. The latent heat released by strong convective rainfall is important for sustaining the ISM circulation (Levermann et al., 2009).

However, it has two opposing effects: on the one hand, the latent heat release in the early stage of an active phase enhances ascending motions by heating the mid-to-lower troposphere (Levermann et al., 2009). On the other hand, latent heat release propagates upward and heats the upper layers of the air column tend to increase the static stability and inhibits further ascending motions (Saha et al., 2012). Also, rainy weather and cloudy skies can also have a cooling effect on the surface in support of suppressing convection (Krishnamurti and Bhalme, 1976). While this thermodynamic perspective is useful to

understand the quasi-biweekly variations of the ISM elements locally, the spatio-temporal variations in the evolution of active and break phases over the Indian monsoon region are known to involve interactions between the wind anomalies and the northward propagation of the major rain band anomalies of the monsoon sub-seasonal oscillation (e.g., (Chattopadhyay et al., 2009; Shige et al., 2017; Wang et al., 2006). Krishnan et al. (2000) hypothesized that the triggering of Rossby waves by suppressed convection over the Bay of Bengal (BOB) initiates ISM breaks through northwest propagation of high

pressure anomalies from the central BOB into northwest India. They noted that the initiation of suppressed convection and anticyclonic anomalies over the equatorial Indian Ocean and central BOB occurred a week prior to the commencement of a monsoon break over India followed by the traversing of suppressed anomalies from the central BOB to northwest India in about 2-3 days.

At sub-seasonal timescales, both tropical and mid-latitude drivers have been proposed to influence the active and break phases

of the ISM rainfall. For example, the Madden-Julian Oscillation (MJO) which governs the sub-seasonal tropical climate variability operating at 30-90 day timescale represents an important tropical driver of the ISM sub-seasonal variability (Zhang, 2005). MJO consists of a transient region of strong convective motions and enhanced precipitation, which propagates eastward from the tropical Indian Ocean to the tropical western Pacific. Normally, only one MJO convective cell is present in these



regions. The MJO influences the ISM system with enhanced convective rainfall activity during strong MJO phases and
negative rainfall anomalies during suppressed MJO phases (Anandh et al., 2018; Mishra et al., 2017; Pai et al., 2011).

Next to tropical drivers, mid-latitude circulation, the North Atlantic variability and mid-latitude wave trains have been also
proposed to modulate the occurrence of active and break phases of ISM (Ding and Wang, 2005, 2007; Kripalani et al.,
1997). A circumglobal teleconnection pattern, characterized by a wave number 5 that encircles the northern hemisphere, has
been associated with anomalous monthly rainfall and surface air temperature across the entire mid-latitude range (Ding and
Wang, 2005). On the one hand, wave trains originating from the North Atlantic may influence the sub-seasonal variability of
the ISM and modulating the intensity of the ISM rainfall (Ding and Wang, 2007; Krishnan et al., 2009). On the other hand,
the diabatic heat sources in association with ISM convection might reinforce the circumglobal wave train propagating
downstream (Ding and Wang, 2005) which is linked to upstream circulation via the monsoon desert mechanism (Cherchi et
al., 2014; Rodwell and Hoskins, 1996). Another recurrent coupled pattern of sub-seasonal variability between mid-latitude
circulation and the ISM is the wave train originating from the north-eastern North Atlantic and propagates with an arch-
shape trajectory through the western Siberian Plains into central Asia (Ding and Wang, 2007). An important feature of this
wave train is the 200 hPa central Asian High which may be able to trigger positive rainfall anomalies over the ISM region.
This wave train generated in the North Atlantic region might also aid in modulating the alternating active and break
conditions over central India (Ding and Wang, 2007; Krishnan et al., 2009; Saeed et al., 2011).

Recent studies have shown that large-amplitude westerly troughs intruding into the Indian subcontinent often coincide with
extreme rainfall events over both the eastern and western Himalayan foothills (Houze Jr et al., 2017; Vellore et al., 2014,
2016), while the wave pattern can impose severe heat wave conditions in the upstream region. An infamous example is the
period of extreme rainfall, which hit the western Himalayan foothills and Pakistan during the end of July and the beginning
of August 2010 together with a Russian heat wave on the European side simultaneously marking the most severe heat waves
in its history on record. These two events have been widely analysed providing compelling evidence of a wave-connection
between the two events (Lau and Kim, 2011).

Plain correlation and composite analysis are commonly used to assess the relationship between two or more climate variables
(Ding and Wang, 2005; Lau and Kim, 2011; Vellore et al., 2014). However, correlation alone cannot distinguish whether the
detected relationships represent actual causal connections or are only spurious links, due to autocorrelation, indirect links or
common drivers. Recently, causal discovery algorithms have been developed and subsequently applied to gain insights into
the physical links of the climate system (Kretschmer et al., 2016; Runge et al., 2015a; Schleussner et al., 2014). One can use
such tools to test whether hypothesized links or teleconnections are likely to represent true physical pathways or rather artefacts
due to spurious correlations. A careful analysis in the context of the ISM thus starts from physical theory and, hence, requires
domain knowledge of ISM dynamics. Using this so-called "theory-guided causal discovery analysis", we here study the two-
way causal links connecting both tropical and mid-latitude regions with the ISM. First, we assess whether the connection



between the ISM and the mid-latitude wave trains can be considered causal in one or both directions. Next, we quantify the relative causal effect of tropical, extra-tropical and internal drivers on the ISM.

## 2 Data and Methods

### 2.1 Data

We define the monsoon trough (MT) region as the region between 18°-25°N and 75°-88°E. We analyse weekly rainfall sums over the MT region from the CPC-NCEP (0.25°x0.25°) observational gridded global rainfall dataset over the period 1979-2016 (Chen et al., 2008) and from the Pai et al. (0.25°x0.25°) observational gridded Indian rainfall dataset over the period 1979-2017 (Pai et al., 2015). In the remainder of this paper, we will mainly focus on the results obtained for the latter data set, while those for the former are provided as parts of the Supplementary Material. Using data taken from the ERA Interim

reanalysis (Dee et al., 2011) for the period 1979-2017, precursor regions are calculated from weekly averaged gridded (1.5°x1.5°) fields including outgoing longwave radiation at the top of the atmosphere (OLR), vertical velocity at 500 hPa (W) and geopotential height at 200 hPa (Z200). The NAO weekly index is obtained by averaging daily data from NOAA (available at ftp://ftp.cpc.ncep.noaa.gov/cwlinks/norm.daily.nao.index.b500101.current.ascii). To identify MJO phases, we use the OLR MJO Index (OMI) provided by NOAA (https://www.esrl.noaa.gov/psd/mjo/mjoindex/). This metric features the first and

second principal components obtained by the empirical orthogonal function (EOF) analysis of OLR in the tropical belt (between 30°N and 30°S) filtered to remove influences outside the MJO time scale (30-90 days). OMI PC2 corresponds to the first principal component of the Real-Time Multivariate MJO index (RMM1), which is widely used in the literature (Kiladis et al., 2014; Pai et al., 2011; Wheeler and Hendon, 2004). All time series of MT rainfall, Z200 and all datasets analysed in this work are detrended and anomalies are calculated at weekly time-steps. Thus, both the climatological and seasonal cycle are

removed.

### 2.2 Causal effect networks

We apply Causal Effect Networks (CEN) and the Response-Guided Causal Precursors Detection (RG-CPD) tool, two recently developed applications of the so-called Peter and Clark (PC) algorithm (Runge et al., 2014; Spirtes et al., 2000).

A CEN detects and visualizes the causal relationships among a set of univariate time series of variables (Kretschmer et al.,

2016). The network is constructed by using the PC-MCI algorithm, a causal discovery algorithm able to distinguish between spurious and true causal links for different time lags of interest (Runge, 2018). Note that the term "causal" rests on several assumptions (Spirtes et al., 2000; Runge, 2018). Here, it should be understood as *causal relative to the set of analysed precursors*, meaning that the identified causal links are valid in the selected set of actors. Adding additional actors may change the structure of the causal network. It is therefore crucial to combine CEN with theoretical domain knowledge (i.e. our "theory-

guided causal discovery analysis" approach).



The PC-MCI algorithm is a two-step algorithm based on a modified version of the PC algorithm (Runge et al., 2014; Spirtes et al., 2000). The first step, the PC-step, identifies the relevant conditions for each variable by iterative independence testing. The second MCI-step, or *momentary conditional independence* step, tests whether the link between two actors can be considered causal. The false discovery rate (FDR) approach as described by Benjamini and Hochberg is applied to correct
inflated p-values due to multiple significance testing (Benjamini and Hochberg, 1995; Benjamini and Yekutieli, 2001). Each step is further described below.

In a variable set $P$ containing $n$ univariate detrended anomaly time series (*actors*), the PC-step identifies the *causal parents* of each element in $P$, among all the remaining elements in $P$. First, the PC algorithm calculates plain correlations between each $i^{th}$ actor in $P$ and all the remaining elements at a certain time lag $\tau$. Those actors that significantly correlate with the $i^{th}$ actor
form the set of its initial parents $P_i^0$ at lag $\tau$. Then, partial correlations between the $i^{th}$ actor and each element in $P_i^0$ are calculated, conditioning first on one condition. If $x$, $y$ and $z$ are elements in $P$, the partial correlation between $x$ and $y$ conditioned on $z$ is calculated by first performing linear regressions of $x$ on $z$ and of $y$ on $z$ and then calculating the correlation between the residuals. If the partial correlation between $x$ and $y$ is still significant at a certain significance threshold $\alpha$, $x$ and $y$ are said to be *conditionally dependent* given variable $z$, i.e., the correlation between $x$ and $y$ cannot be (exclusively) explained by the
influence of variable $z$. When the opposite happens, the link is thus spurious and therefore filtered out, and $x$ and $y$ are called *conditionally independent*. This process leads to a reduced set of parents $P_i^1$. In the next step, the process is repeated but conditioning on two conditions, leading to a second set of parents $P_i^2$. The algorithm converges when the number of causal parents contained in $P_i^n$ is equal or greater than the number of conditions needed to calculate the next partial correlation. At the end of the PC-step, each element in $P$ has its own set of parents, which then enters the MCI-step. In the MCI-step, the
partial correlation between an actor and its set of parents is calculated again but conditioned simultaneously on both the sets of parents of the $i^{th}$ actor and the sets of parents of each of the parents of the $i^{th}$ actor. Those parents that pass the MCI test will then form the final set of parents for the $i^{th}$ actor (Runge et al., 2017). A numerical example of these steps is given in the SI, Text S1.

**2.3 Response-guided causal precursor detection**

RG-CPD identifies the causal precursors of a response variable based on multivariate gridded observational data (Kretschmer et al., 2017). It combines a response-guided detection step (Bello et al., 2015) with the PC-MCI causal discovery step (Runge et al., 2014; 2015a; 2015b; Spirtes et al., 2000). Without requiring an a priori definition of the possible precursors, RG-CPD searches for correlated precursor regions of a variable of interest (i.e. the *response variable*) in multivariate gridded data and then detects causal precursors by filtering out spurious links due to common drivers, autocorrelation effects or indirect links.
Using correlation maps, an initial set of precursors is identified in relevant meteorological fields by finding regions that precede changes in the response variable at some lead time. Adjacent grid points with a significant correlation of the same sign at a level of $\alpha=0.05$ are spatially averaged to create single one-dimensional time series, called *precursor region* (Willink et al.,



2017). The variance inflation due to serial correlations is again addressed using the Benjamini and Hochberg FDR correction (Benjamini and Hochberg, 1995; Benjamini and Yekutieli, 2001). In the second step, PC-MCI identifies the set of *causal*
*precursors* for the response variable. The results are presented as precursor regions on a global map that are identified to be causally linked with the response variable (ISM rainfall).

## 3 Results

### 3.1 Causal testing of the two-way ISM-circumglobal teleconnection mechanism

First, we assess whether the two-way interaction between the circumglobal wave train (that characterizes the boreal summer
circulation) and the ISM rainfall, as hypothesized by Ding & Wang (2005, hereafter DW2005), is reproduced using CEN. We will refer to this theory as the monsoon-circumglobal teleconnection mechanism.

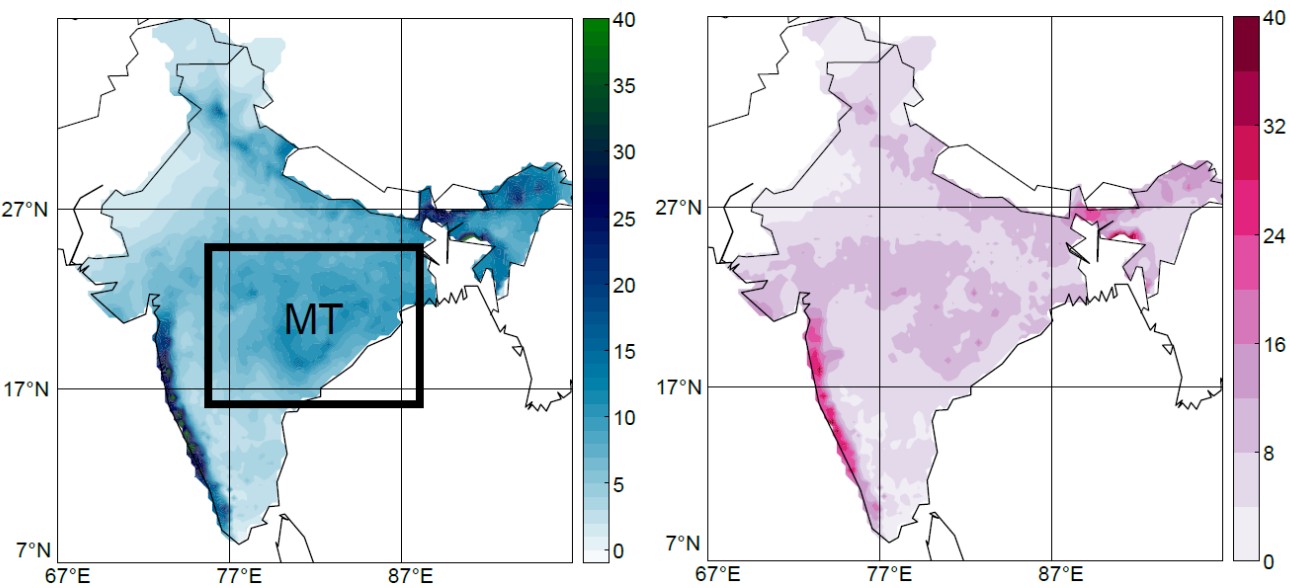

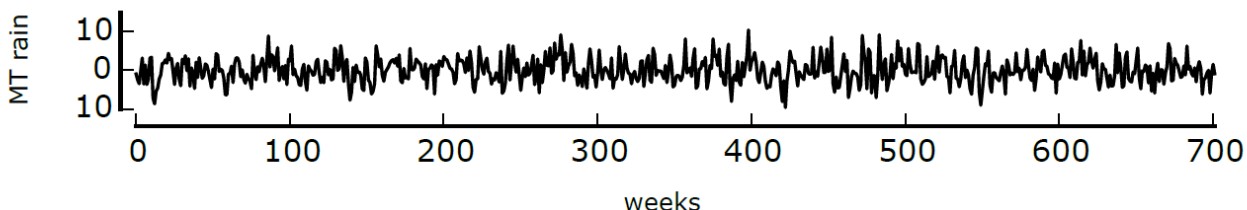



**Figure 1. Rainfall climatology over India.** Panel (a): JJAS rainfall climatology over the 1979-2017 period from the Pai et al. dataset. The black box identifies the MT region. Panel (b): standard deviation (s.d.) of weekly JJAS rainfall over the 1979-2017 period from the Pai et al. dataset. Panel (c): time series of weekly MT JJAS rainfall over the period 1979-2017; each year contains 18 weeks, with the first week starting on the 27th of May.

Figures 1a,b show the JJAS climatology and the standard deviation (s.d) of weekly ISM rainfall from Pai et al. dataset for the period 1979-2017. We average the rainfall over the MT region and identify a univariate time series, which represents the weekly variation of the ISM during JJAS over the MT region (Fig. 1c). This time series contains 18 weeks for each of the 39 analysed years, each year starting on the 27th of May. The analysis of the MT rainfall starts on the 3rd week (10th of June) for a total of 16 time slots per year. The first two time slots allow detecting lagged relationships, and the PC-MCI algorithm requires to add twice the maximum time lag explored (here a maximum lag of 1 week is chosen). Thus, information from the previous year does not interfere with the following year. The weekly time scale is considered to represent the relevant interaction between ISM rainfall and Northern Hemisphere atmospheric circulation at a sub-seasonal timescale (Ding and Wang, 2007; Krishnamurti and Bhalme, 1976; Vellore et al., 2014).







**Figure 2. Mid-latitude variability associated with the ISM.** Panel (a) and (b): EOF1 and EOF2 for the JJAS weekly Z200 field in the northern mid-latitudes for the period 1979-2017. Panel (c): correlation between weekly MT rainfall and Z200 (lag = -1 week). Panel (d): the CGTI region (white box) and the correlation between CGTI and Z200 (lag = 0), which forms the circumglobal teleconnection pattern. In



panels c,d correlation coefficients and anomalies with a p-value of $p < 0.05$ (accounting for the effect of serial correlations) are shown by black contours. Panel (e): temperature anomalies over the Northern Hemisphere during weeks with CGTI > 1 CGTI$_{std}$ minus weeks with CGTI < -1 CGTI$_{std}$. Panel (f): as panel (e) but for rainfall anomalies from the CPC-NCEP dataset for the period 1979-2017. In panels e,f

anomalies with a p-value of $p < 0.05$ (accounting for the effect of serial correlations) are shown by black contours, while grid points significant with non-corrected p values are shade.

Following DW2005, we calculate the first and second empirical orthogonal functions (EOFs) of weekly averaged Z200 fields over the Northern Hemisphere (0°-90°N) for the summer months coinciding with the ISM season (June to September, weeks

23 to 38). Figures 2a,b show the first and second EOFs. EOF1 represents the dominant component of sub-seasonal variability of the Z200 field, and qualitatively resembles the s.d. of Z200 (see Fig. S1 in the SI). The EOF2 represents the second dominant pattern of sub-seasonal variability of the Z200 field. Next, we test whether it is linked to the circumglobal teleconnection pattern, as suggested in DW2005.

Figure 2c shows the spatial correlation structure of MT rainfall with variability of Z200 in the Northern Hemisphere at weekly

timescale with Z200 leading the MT rainfall by one week (lag = -1). Following DW2005, we define a circumglobal teleconnection index (CGTI) as the weekly-mean Z200 spatially averaged over the region 35°-40°N, 60°-70°E (white box in Fig. 2d). The contemporaneous (lag=0) spatial correlation structure between the CGTI and Z200, i.e. the circumglobal teleconnection pattern, is shown in Fig. 2d. A large region of strong positive correlation surrounds the Caspian Sea, while downstream, i.e. moving from west to east following the mid-latitude westerlies, a circumglobal wave train is shown with 4

positive centres of action positioned over east Asia, the North Pacific, North America and western Europe. Despite different temporal averaging and using different datasets, our results with 5 centres of positive correlation strongly resemble the circumglobal teleconnection pattern described by DW2005, in terms of sign and the geographical position of its centres of action. In both Fig. 2c and 2d correlation values with corrected p-values $p < 0.05$ are highlighted with black contours. Correlation values are calculated with a two-sided p-value for a hypothesis test whose null hypothesis is that there is no

correlation, using the Wald Test with a t-distribution of the test statistic. All p-values are corrected using the Benjamini and Hochberg FDR correction to address the variance inflation due to serial correlations (Benjamini and Hochberg, 1995; Benjamini and Yekutieli, 2001).

The circumglobal teleconnection pattern (Fig. 2d) shows a weak negative spatial correlation with the EOF1 pattern ($r = -0.03 \pm 0.02$, confidence interval at $\alpha = 0.05$). The spatial correlation of EOF2 (Fig. 2b) with the circumglobal teleconnection pattern

(Fig. 2d) at a global scale is $r = 0.29 \pm 0.01$, but the spatial correlation with the circumglobal teleconnection pattern increases over the Eurasian sector ($r = 0.42 \pm 0.02$). Contrarily, the spatial correlation with EOF1 and the circumglobal teleconnection is also low when only the Eurasian sector is taken into account ($r = 0.07 \pm 0.03$). When EOFs are calculated only on the Eurasia sector (0°-90°N – 0°-150°E), the order of the first and second EOFs is reversed, but the spatial patterns are very similar: EOF1$_{Eurasia}$ is strongly spatially correlated with EOF2 ($r = 0.95 \pm 0.01$) and with the circumglobal teleconnection pattern ($r =$

$0.44 \pm 0.02$; see SI, Fig. S2). Thus, the choice of the region to calculate the EOFs does not strongly affect the results.





The time series for the principal component of EOF2 also significantly correlates with the CGTI time series ($r = 0.29 \pm 0.01$) while the correlation for the first principal component is low and not significant ($r = -0.06 \pm 0.08$). The circumglobal teleconnection pattern also strongly resembles the correlation structure between MT rainfall and Z200 at lag -1 (Fig. 2c) with a spatial correlation of $0.65 \pm 0.01$. These findings are consistent with those of DW2005, and thus illustrate a likely interaction
between MT rainfall and Z200 variability.

We also calculate 2m-temperature and precipitation anomalies of the composite of weeks with CGTI > 1 CGTI$_{std}$ minus composites of weeks with CGTI < -1 CGTI$_{std}$ (where CGTI$_{std}$ is the s.d. of the CGTI index). Figures 2e,f present the corresponding 2m-temperature and precipitation anomalies from the CPC-NCEP dataset, respectively, showing anomalies with p-values $p < 0.05$ shaded and highlighting anomalies that have corrected p-values $p < 0.05$ by black contours. In both
variables, a wave train from Western Europe to India via European Russia is detected. Wet and cold anomalies appear over central India and European Russia, while warm and dry anomalies are found over Western Europe and east of the Caspian Sea. Warm and dry anomalies appear together, however precipitation anomalies show a slight eastward shift with respect to temperature anomalies. Precipitation anomalies are weaker than those found for 2m-temperature, however a clearly defined wave pattern appears over the Eurasian sector.

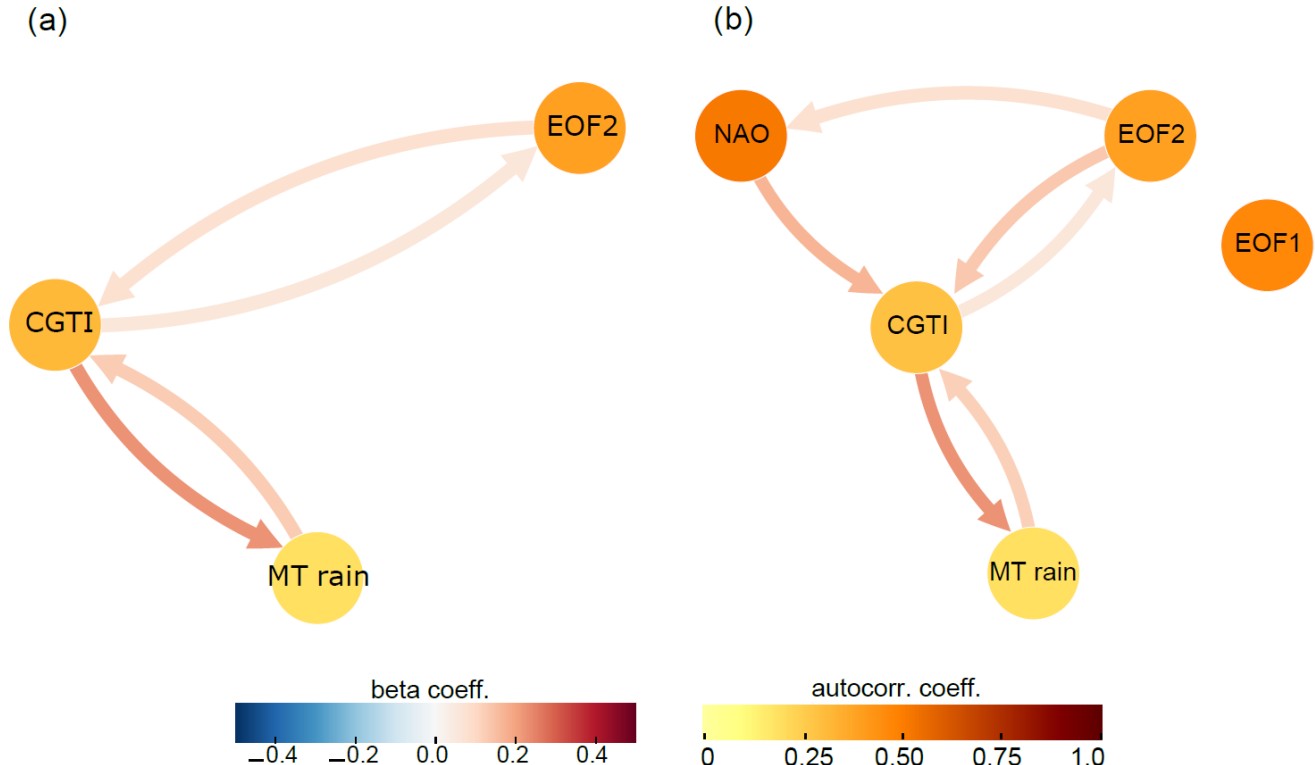


**Figure 3. Causal mid-latitude interactions with the ISM.** Panel (a): Causal Effect Network (CEN) built with CGTI, the PC of EOF2 and MT rainfall for the period 1979-2017. Panel (b): as panel (a) but with the addition of the PC of EOF1 and NAO. The strength of causal links



expressed by the standardized regression (beta) coefficients and autocorrelation coefficients are shown. All links have a lag of 1 week. See the main text for discussion.


Following the DW2005 hypothesis, we build a Causal Effect Network (CEN) with CGTI, MT rainfall and the time series for the principal component of EOF2 (Fig. 3a). This CEN depicts a positive two-way connection both between CGTI and MT rainfall and between CGTI and EOF2. This implies that anomalously high CGTI values (with relatively high Z200 east of the Caspian Sea) enhance MT rainfall at a 1-week lag, while lower CGTI values have the opposite effect. The link directed from

MT rainfall to CGTI illustrates a reverse influence, creating a positive feedback between CGTI and MT rainfall, consistent with DW2005. The strength of the causal link between the CGTI and MT rainfall is expressed by the path coefficients, i.e. the "expected change in $X^j$ (in units of its s.d. and relative to the unperturbed regime) at time $t$ if $X^i$ was perturbed at time $t–\tau$ by a one s.d. delta peak" (Runge et al., 2015a). Here, due to the fact that only lags at $\tau=1$ are accounted for, the path coefficients also correspond to the total causal effect (CE).

The causal link strength of the CGTI acting on the MT rainfall is $\beta_{CGTI \to MT}$ ~0.2 (meaning that a change of 1 s.d. in CGTI leads to a change in 0.2 s.d in MT rainfall under the conditions mentioned in the previous paragraph), while the reverse link is weaker ($\beta_{MT \to CGTI}$~0.1) but still significant. EOF2 shows a two-way link with the CGTI. The links between EOF2 and CGTI support the DW2005 hypothesis that wave trains in the mid-latitudes (represented by EOF2) affect the MT rainfall via the CGTI. Ding and Wang (2007, hereafter DW2007) formulated a hypothesis involving the existence of a wave train from the North

Atlantic down to the Indian Monsoon domain. To assess whether the North Atlantic variability affects the MT rainfall, we add the NAO index to our original CEN. In order to check whether the first mode of variability in the Northern Hemisphere may also play a role in shaping MT rainfall variability, we additionally include EOF1. Figure 3b shows the resulting CEN: the causal links identified in the previous CEN (Fig. 3a) remain unaltered, and two additional positive links from NAO to CGTI and from the EOF2 to the NAO emerge. A positive NAO phase will strengthen the CGTI at 1-week lag ($\beta_{NAO \to CGTI}$~0.2) with

a reverse influence from EOF2 on NAO ($\beta_{EOF2 \to NAO}$~0.1), though weaker. EOF1 does not show a causal connection with CGTI or with any other actor. This CEN unveils both an influence of the mid-latitude atmospheric dynamics (EOF2) and North Atlantic variability (NAO) on MT rainfall and a backward link from the MT rainfall to EOF2 via CGTI, thus supporting both the DW2005 and DW2007 hypotheses.

### 3.2 Regional extratropical features affecting the Indian summer monsoon

The EOF-based CEN analysis depicts the total hemispheric response of Z200 without differentiating among the influences of different geographical regions. To detect influential regions, we apply RG-CPD to search for causal precursors of both the CGTI and MT rainfall at 1-week lead time in weekly OLR and Z200 fields. In the tropical belt, OLR is often used as a proxy of rainfall and convective activity due to its relation with the temperature at the top of the clouds. Deep convection is characterized by high altitude cloud tops and low emission temperatures (and thus low OLR values), while with clear sky

conditions, the emission temperature of the land surface is higher and leads to larger OLR values (Krishnan et al., 2000).



Figure 4 summarizes these findings. The right column (Fig. 4) shows the MT region and represents lag 0. Moving towards the left, the second column shows the correlation maps (top) and causal precursors (bottom) of the MT rainfall identified at 1-week lead time in Z200 (Fig. 4c). Causal precursors describe an arch-shaped wave train from Western Europe to India. The wave train features one low-pressure region (L1 over European Russia) and two high-pressure regions (the CGTI and H1 over

Western Europe). The leftmost column in Fig. 4 shows the correlation maps (top) and causal precursors (bottom) of the CGTI identified in the Z200 (Fig. 4a) and OLR (Fig. 4b) fields at 1-week lead time with respect to the CGTI (2-week lead time with respect to the MT rainfall). Again, both Z200 and OLR correlation maps show an arch-shaped wave pattern emanating from the North Atlantic and propagating towards the Caspian Sea via European Russia. The associated Z200 causal precursors for the CGTI clearly depict this mid-latitude wave train both in Z200 and OLR (Figs. 4a,b). The wave train features two lows (L1

over European Russia and L2 over the eastern North Atlantic) and two highs (the CGTI and H1 over Western Europe). OLR causal precursors (Fig. 4b) depict only the L1 and H1 components of the wave, as the correlation over L2 is not significant. Moreover, the prominent influence of the tropical belt on the CGTI is also detected (OLR1, Fig. 4b). This result further supports the hypothesis that a wave train coming from the mid-latitudes influences the ISM circulation system via the CGTI region as already shown in Fig. 2. Moreover, temperature and precipitation anomalies shown in Figs. 2e,f strongly correspond to the

regions H1, L1 and CGTI identified in Figs. 4a,c.





**Figure 4. Mid-latitude causal precursors of ISM.** Panel (a): correlation of CGTI with Z200 at 1-week lead time (top), and the causal precursors of CGTI identified via RG-CPD (bottom) for the period 1979-2017. Panel (b): as for panel (a) but for OLR fields. Panel (c): correlation map for weekly MT rainfall and Z200 field at 1-week lead time (top) and the causal precursors identified via RG-CPD (bottom). Panel (d): ISM rainfall over the MT region from the Pai et al. dataset (reproduced from Fig. 1a).

The CEN built with MT rainfall, CGTI and the upstream part of the mid-latitude wave train, i.e. L1 and H1, is shown in Fig. 5. The causal links between the CGTI and the MT rainfall remain unaltered (see Fig. 3), with the CGTI mediating the connection between the mid-latitude wave (represented by H1 and L1) and the MT rainfall. Further, the obtained CEN can be interpreted as a wave train that propagates downstream from the east Atlantic towards the monsoon region: when H1 increases, the CGTI also increases with 1-week lag, while when L1 deepens, CGTI increases. In both cases, this reflects an eastward propagation of a wave from Western Europe via Russia towards the CGTI region. The connections from the CGTI back to H1 and from L1 back to H1 have opposite signs as compared to the downstream links. Thus deepening L1 strengthens the CGTI downstream, but weakens H1 upstream, which is an expression of the eastward propagating wave. A consistent result is found





when both L1 and L2 are included in the CEN, although in this case only the downstream causal links are detected (see SI,

Fig. S3). In general, this result further confirms the hypothesis proposed by DW2007 that a wave train coming from the North

Atlantic influences MT rainfall at about 2-week lead time.

**Figure 5. Mid-latitude wave train.** CEN built with the MT rainfall, CGTI, L1 and H1 (as identified in Fig. 4a) for the period 1979-2017.

The strength of causal links expressed by the standardized regression (beta) coefficients and autocorrelation coefficients are shown. All links

have a lag of 1 week. See the main text for discussion.



### 3.3 Internal feedbacks in the monsoon circulation

Next, we perform a similar analysis using fields of vertical velocities (W) and OLR to capture the internal feedbacks and

dynamics of the ISM convective updraft. Their correlation maps and detected causal precursors are shown in Figs. 6a,b. MT rainfall has four causal precursors in OLR. A large region located over India and the Maritime Continent (OLR1, Fig. 6a) shows a negative causal link. A region covering parts of the Himalayan plateau and extending toward the Arabian Peninsula shows a positive causal link (OLR2, Fig. 6a). A third region is found north of the Caspian Sea, over western Russia (OLR3) with a negative causal relationship with MT rainfall. Finally, a last region located over northcentral Europe showing a negative

causal relationship with MT rainfall is found (OLR4, Fig. 6a). OLR1 spatially overlaps with W1, the largest causal precursor in the vertical wind field, representing the summer branch of the ITCZ over the northern Indian Ocean and western Pacific Ocean (Fig. 6b, top panel). The positive correlation of W1 and negative correlation of OLR1 with MT rainfall indicate that ascending motions and associated high-level cloud formation (reducing OLR) are followed by enhanced rainfall over the MT region with a lag of 1 week. OLR2, OLR3 and OLR4 largely overlap with regions H1, L1 and the CGTI region identified in

Fig. 4c, further supporting the importance of this mid-latitude wave pattern in modulating the rainfall over the MT region.





**Figure 6. Tropical causal interactions of ISM.** Panel (a) shows the correlation map for weekly MT rainfall and the global OLR field at 1-week lead time (top panel) and the causal precursors identified via RG-CPD (bottom panel) for the period 1979-2017. Panels (b): as for panel (a) but for W fields. Panel (c) and (d) show the CENs build with W1, OLR1 and MT rainfall and MT rainfall, W1, CGTI and MJO2, respectively. The strength of causal links expressed by the standardized regression (beta) coefficients and autocorrelation coefficients are shown. All links have a lag of 1 week. See the main text for discussion.



Figures 6c,d show two CENs constructed from the MT rainfall causal precursors in OLR and W (region OLR1 as defined in
Fig. 6a and W1 as defined in Fig. 6b), and including CGTI and MT rainfall itself. Since the OLR and W fields are more noisy
than Z200, we use only the two dominant causal precursors. Although the regions OLR1 and W1 show a large spatial overlap,
they represent two different components of the ISM system. OLR is calculated at the top of the atmosphere and low OLR
values represent high-clouds that reach the top of the troposphere. Thus, OLR is a proxy for deep convection and clouds. W is
calculated at 500 hPa and W1 thus represents the ascending branch of the ISM circulation cell and nearby ITCZ. The CEN
built with OLR1, W1 and the MT rainfall at weekly timescale (Fig. 6c) represents the internal dynamics of the monsoon cell.
Enhanced vertical motions (W1) precede an increase in the MT rainfall by one week, while stronger MT rainfall leads to
weaker ascending motions one week later. Meanwhile, more clouds (lower OLR1) are associated with enhanced W1 one week
later (thus enhancing the ISM circulation). Contrarily, enhanced rainfall shows a negative feedback on both OLR1 and W1:
stronger MT rainfall leads to reduced vertical motions and clearer skies (lower W1 and higher OLR1, respectively). This CEN
thus depicts a negative feedback intrinsic to the ISM internal variability: while enhanced ascending motions and thus
strengthened ISM circulation lead to stronger MT rainfall, enhanced rainfall leads to weaker ISM circulation and in turn
diminished MT rainfall. From a physical point of view, this can be explained via an increase of atmospheric static stability due
to latent heat release in the higher layers of the troposphere. This mechanism is shown in Fig. S4 (see SI), where we build a
CEN with MT rainfall, sea level pressure (SLP) over the Bay of Bengal (SLP_BOB), temperature at 400hPa (Tp4_MT) and
temperature at 600 hPa (Tp6_MT). The resulting causal links show that while a decrease in SLP over the Bay of Bengal is
followed by an increase in MT rainfall and both Tp4_MT and Tp6_MT, an increase in Tp4_MT leads to a decrease in MT
rainfall and increase in SLP_BOB, which is in turn conductive for decreased MT rainfall. The described mechanism is in
agreement with what was proposed by Saha et al. (2012) and Krishnamurti and Bhalme (1976).

Next, we test the causal relationships identified between MT rainfall and W1 when adding the CGTI and MJO variability to
the CEN (Fig. 6d). MJO variability is expressed by the OMI PC2 index (here referred to as MJO2), the second EOF of OLR
in the tropical belt (Kiladis et al., 2014).  OMI PC2 corresponds to the RMM1 index, widely used in previous work on the
relationship between the MJO and the ISM (Kiladis et al., 2014; Mishra et al., 2017; Pai et al., 2011). Positive RMM1 values
correspond to MJO phases 3-6 (also referred to as the active phases of MJO) and physically represent the presence of strong
convection activity propagating eastward from the Indian Ocean towards the western Pacific. Several studies show that MT
rainfall is enhanced during the active MJO phase (Anandh et al., 2018; Mishra et al., 2017; Pai et al., 2011). In this CEN, the
links between the MT rainfall and W1 remain unaltered, while the CGTI shows a positive feedback with W1: an increase in
the CGTI causes stronger ascending motions, while stronger ascending motion over the Indian region strengthens the CGTI.
The CGTI has a direct positive causal link to MT rainfall (Fig. 6d). Even though the direct link from MT rainfall to CGTI has
now disappeared (likely because this link was already relatively weak in a simpler CEN configuration, see Fig. 3a), the link
between the ISM circulation and the mid-latitudes remains via W1. Stronger ISM circulation (higher W1) leads to increased
CGTI and vice versa as seen in Fig. 3, where higher CGTI leads to enhanced MT rainfall and vice versa. MJO2 displays a



positive causal link with W1, meaning that OMI PC2 positive values lead to enhanced vertical motions with 1-week lead time and, as a consequence, enhanced MT rainfall with 2-week lead time. However, stronger W1 leads to decreased MJO2 (negative causal link from W1 to MJO2). Thus, MJO2 shows exactly the same causal relationships (but opposite signs) with W1 as

shown for OLR1 in Fig. 6c, likely because the OMI index is also defined based upon OLR fields in the tropical belt and the OLR pattern shown by the second EOF of the OMI index largely overlaps with our OLR1 region (Kiladis et al., 2014).

### 3.4: Combining local, tropical and mid-latitude causal interactions

Finally, we bring together the findings obtained with CEN and RG-CPD throughout this study and summarize them in a single CEN to provide an overall picture and test the consistency of the results. We include the most important identified regions

from both the tropics and the mid-latitudes together in a single CEN (Fig. 7). Specifically, this CEN is built with elements that come from both, the DW2005 and DW2007 hypotheses (Fig. 3) and from our RG-CPD analysis (Figs. 4 and 6). Our results show that the influence of both, the mid-latitude circulation (EOF2) and the North Atlantic (NAO), on the MT rainfall is mediated via the CGTI and is robust: the structure and the direction of the causal links are retained. The backward influence of the MT rainfall on the mid-latitude circulation is weaker and more complex, as shown in Fig. 6d. CGTI has both a direct

causal link to MT rainfall and an indirect one going via W1.





**Figure 7. Combined mid-latitude and tropical causal interactions of ISM.** CEN built with W1, MJO2, MT rainfall, NAO, CGTI and EOF2 for the period 1979-2017. The strength of causal links expressed by the standardized regression (beta) coefficients and autocorrelation coefficients are shown. All links have a lag of 1 week. See the main text for discussion.




MT rainfall and W1 show a negative feedback, with increased W1 leading to enhanced MT rainfall but stronger MT rainfall leading to weaker W1. The MT rainfall is also influenced by MJO2 via W1 with a one-way connection. Suppressed or weakened ascending motions promote lower MJO2 one week after, thus promoting the switch toward MJO phases 7-8 and 1-

2, also known as suppressed MJO phases (see SI, Fig. S11). Suppressed MJO phases are in turn linked with the onset of break phases of the Indian monsoon (Pai et al., 2011).

To quantify the relative influence of tropical and mid-latitude teleconnections on MT rainfall, we report the causal effect strength of each link, which is given by the beta coefficient of the multilinear regression of standardized MT rainfall with the identified causal parents. W1 has the strongest causal effect on MT rainfall with $\beta_{W1\rightarrow MT} = 0.53$, implying that a one standard

deviation shift in W1 causes about a half standard deviation change in MT rainfall after one week (under the previously mentioned conditions, see above). The CGTI influences MT rainfall directly and indirectly via W1 and the total causal effect is given by $\beta_{CGTI\rightarrow MT} + \beta_{CGTI\rightarrow W1} \cdot \beta_{W1\rightarrow MT} = 0.26$. The causal effect of CGTI is thus roughly half as strong as that of the internal variability of the Indian monsoon system as represented here by W1. MJO2 has a causal effect on W1 of $\beta_{MJO2\rightarrow W1} = 0.29$ and is influenced by W1 with $\beta_{W1\rightarrow MJO2} = -0.36$. Thus, taking both W1 and MT as representatives of internal ISM

dynamics, the effects of external drivers from both tropics and mid-latitudes on this internal dynamics are of similar magnitude, which is about half as strong as that of W1 on MT. Looking more specifically at the causal effect of MJO2 on MT rainfall mediated via W1, we find $\beta_{MJO2\rightarrow W1}*\beta_{W1\rightarrow MT} = 0.15$, i.e., the tropical driver effect on MT rainfall itself is only about half as strong as that of CGTI as the key extratropical driver. Beta coefficient for all the identified causal links are reported in Table S1 (see SI).

Moreover, we calculate the average causal effect (ACE) and average causal susceptibility (ACS) for each actor. While ACE gives a measure of the causal effect that each actor has on the rest of the network, ACS measures the sensitivity of each actor to perturbations entering in any other part of the network (Runge et al., 2015a). In this CEN, W1 has the highest ACE (~0.21) and CGTI the second highest ACE (~0.08). The MT rainfall shows the strongest ACS (~0.14), followed by W1 with ~0.13. ACE and ACS values for each actor are further summarized in Table S2, see SI. These values again stress the importance of

both, ISM internal variability and CGTI, in mediating mid-latitude waves towards the ISM.

We performed the same analysis using the CPC-NCEP rainfall dataset over the period 1979-2016 (see SI, Figs. S5-S11) and find that our results are robust when using this different rainfall dataset. All causal links are reproduced with the same directions and only the magnitudes of the causal effects show minor changes.

## 4. Discussion and conclusions

In this study, we apply causal discovery algorithms to analyse the influence of global middle and upper tropospheric fields on the ISM rainfall and study the two-way causal links between the mid-latitude circulation and ISM rainfall. We perform a validation of both the monsoon-circumglobal teleconnection hypothesis proposed by DW2005 and the North Atlantic-monsoon connection proposed by DW2007 using causal discovery tools. We use RG-CPD to detect causal precursors from



both mid-latitude and tropical regions and then apply CEN to assess the influence of different drivers of MT rainfall and their

relative contribution to the MT rainfall sub-seasonal variability.

Our causal analyses confirm the influence of the mid-latitude circulation on MT rainfall via the CGTI, as hypothesized by DW2005. We also confirm that MT rainfall forces the mid-latitude circulation via the CGTI but this link is weaker. Moreover, we use a causal precursor identification tool and find a wave train that emanates from the eastern Atlantic stretching towards India. This wave pattern is visible in geopotential height fields, temperature and precipitation anomalies, and acts on MT

rainfall via the CGTI with a 1-2 week lead time, in agreement with the DW2007 hypothesis.

Our results show that RG-CPD can detect the well-known circulation features of the MT rainfall with 1-week lead time, without using any a priori theoretical or geographical constraint to select the causal precursors among all precursor regions demonstrating the efficacy of the presented method. Moreover, causal discovery tools can quantify the causal influence of tropical drivers versus extra-tropical influences and internal dynamics of the ISM sub-seasonal circulation dynamics.

Adding internal dynamics helps to further understand the mechanisms that underlie this mid-latitude-ISM relationship. Internal variability of the ISM system has the strongest effect on MT rainfall. The influence from the extratropics on MT rainfall, as mitigated by CGTI, is about half of the magnitude of that of internal dynamics, while the influence of MJO as the key tropical driver on MT rainfall is just about one fourth (Fig. 7). However, when taking MT rainfall and vertical wind field over the Indian subcontinent together as two interdependent yet different facets of the ISM, we find that the general effect of tropical

drivers on the system is of a similar order of magnitude as that of the extratropical drivers, while looking on the one hand on the effect of MJO on the circulation and on the other hand on the total effect of CGTI on MT rainfall via both, directed linkages and through a parallel influence on the vertical wind field over India.

The reported findings are in good agreement with the existing literature. It is well known that internal variability dominates ISM inter-annual variability (Goswami and Xavier, 2005). Our causal approach enables to quantify the relative importance

of local internal dynamics, separate it from the influence of remote actors, and remove spurious factors. The negative feedback in the ISM internal variability, here represented by the opposite relationship between MT rainfall and W1 (see Figs. 6d and 7), further supports the hypothesis that the internal dynamics of the ISM itself provides a mechanism to switch from an active to a break phase. The physical mechanism can include both radiative effects (Krishnamurti and Bhalme, 1976) and local changes in static stability due to the latent heat release that follows convective precipitation (Saha et al., 2012). While

strong upward motions precede strong MT rainfall, enhanced rainfall over the ITCZ region is known to lead the initiation of breaks by 7-10 days (Krishnan et al., 2000). Moreover, suppressed convection over the Bay of Bengal initiated over the tropical Indian Ocean and associated westward propagating Rossby waves bring break conditions over the monsoon trough (Krishnan et al., 2000).

Moreover, our results also support previous findings that suggest a link between the active MJO phases (3-6, corresponding

to positive RMM1 values) and enhanced ascending motions over the MT region and adjacent Maritime Continent which in turn promote enhanced MT rainfall (Fig. 6d and 7) (Anandh et al., 2018; Mishra et al., 2017; Pai et al., 2011).

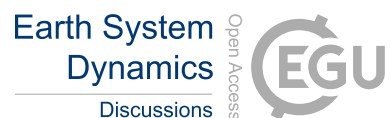

Our theory-guided causal effect network approach, i.e. creating CENs starting from physical hypotheses, enables us to: (1) test those hypotheses in a causal framework, removing the influence from spurious correlations, and (2) quantify the relative strength (i.e. causal effect) of different local and remote actors. With this approach, one can gain insight in what role each part

of a complex system such as the ISM circulation plays in relation to the other components. However, domain knowledge is essential to be guided by known physical processes and associated timescales. By combining RG-CPD and CEN, one can test initial hypotheses and perform further more explorative causal analyses to identify new features. For example, in this study, we first identify our initial actors based on the literature. Then, we increase the pool of actors by searching for causal precursors using RG-CPD. Finally, we reconstruct a CEN that combines those findings and helps to put them into a broader context. This

approach can be applied to both observational data (as done here) and climate model data to validate the underlying processes behind sub-seasonal variability, which might pave the way for improved forecasts.

The described identification and quantification of causal dependences is based on linear statistical models between the different considered variables quantified in terms of partial correlations. While such linear models can provide useful approximations of real-world climate processes, there could be cases in which they miss other existing linkages that are not described by linear

functional relationships. In turn, extending the present analysis to a fully nonlinear treatment is straightforward but would come on the cost of much higher computational demands, which is why we have restricted ourselves in this work to the linear case. Nevertheless, accounting for possible nonlinearities may add further information on the inferred mechanisms and should therefore be undertaken in future research.

Our results indicate that, on weekly timescales, the strength of the influence from the mid-latitude on MT rainfall itself is about

twice as large as that from the tropics (MJO) but about a factor two smaller than the ISM internal dynamics. However, the tropical (MJO) effect on the associated vertical wind speed over the MT region is of about the same magnitude as that of extratropical drivers on MT rainfall. Related to the confirmed relevance of extratropical drivers for ISM variability at weekly scales, we emphasise there exists a substantial body of literature suggesting that the influence from the mid-latitudes is particularly important for extremes (Lau and Kim, 2011; Vellore et al., 2014, 2016). Future work should therefore aim to

further disentangle the specific mechanisms that particularly act in the context of extremes.

**Acknowledgments**

We thank ECMWF and NCEP for making the ERA-Interim and CPC data available. This work was supported by the German Federal Ministry of Education and Research (projects GOTHAM, Sacre-X and CoSy-CC2). Code for the causal discovery algorithm is freely available as part of the Tigramite Python software package at https://github.com/jakobrunge/tigramite.

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
