# Peer review of "Tropical and mid-latitude teleconnections interacting with the Indian summer monsoon rainfall: A Theory-Guided Causal Effect Network approach"

_Earth System Dynamics, 2019_

## Referee Comment (RC1) · Anonymous Referee #1 · 3 May 2019

REF: ESD-2019-11

TITLE: Tropical and mid-latitude teleconnections interacting with the Indian summer monsoon rainfall: A theory-guided causal effect network approach AUTHORS: Di Capua G, Kretchmer M, Donne RV, van den Hurk B, Vellore R, Krishnan R, Coumou D

In this study the authors want to identify causal effect networks for the ISM and tropical and extra-tropical regions. According to the authors the causal network applied is able to test physical hypothesis, to explore for causal links and to quantify relative

contributions. The idea of the analysis is interesting and the potential for the results is high, nevertheless I don't see much of what claimed in the present analysis. The main problem with this manuscript is that it is really hard to read: there are clear limitations due to the large numbers of acronyms in the text, hard to remember and that require back and forth from the results sections to the methodology section. In terms of results, I can see the verification of the results as found in previous works cited, but I hardly see here new results and a quantification of the relative role of influence on the MT rainfall. In my opinion revisions is needed before the work can be accepted for publication in the journal. Below main and minor comments are listed in details.

Main comments: 1 - the manuscript is really hard to read because of too many and complex acronyms. I think the authors need to find a way to simplify the reading using nick-names for the tools applied instead of rude acronyms difficult to remember; 2 - the list of the new findings using these tools should be clearly highlighted in the manuscript, as well as the different weights of the different precursors considered. These capabilities of the tools used are claimed also in the abstract but the results are not clearly extrapolated and summarized in the text; 3 - Fig 2e,f: how do you explain the propagation in t2m and precipitation? what about winds? 4 - Figs 3,5,6 and 7: in these type of figures arrows indicate the intensity of the beta coefficients, while the color of the circle the auto-correlations: how are these information combined in interpreting the results? Also what is the real meaning of the intensity of the beta coefficient. It seems in most of the case quite small, thus indicating a very small relationship (?), and it is large only in the case of linking W1 to MT rain (Fig 7) and in MJO2 linking W1 (Fig 7). How are these measures able to weight for the different factors influencing MT rainfall?

Minor comments: 1 - Line 43: MJO first referenced but without expanding acronym; 2 - End of Introduction: the content of the manuscript (what in sect 2, sect 3 and so on) should be detailed at the end of the Introduction to simplify reading; 3 - Line 156: "conditioning first on one condition", quite uneasy sentence to read; 4 - Line 197: how do you have chosen those dates? 5 - Line 242: I don't understand how if -0.06 is

non-significant, the value -0.03 related to EOF1 (previous line 233) can be significant (statistically?) 6 - Line 271: likely worth repeating what conditions you are exactly referring to; 7 - Fig 5: the intensity of the link for this figure is not done in the text, as instead it was done for Fig 3. Why this difference in the treatment of the interpretation of the same kind of figure? 8 - Line 329: "internal" to what? what does this mean exactly? 9 - Lines 378-379: I don't understand how/why this differs from what stated before for Fig 3.

---

## Referee Comment (RC2) · Anonymous Referee #2 · 18 May 2019

This study attempted to identify and quantify the causal relationships between tropical/mid-latitude precursors and ISM rainfall on sub-seasonal timescale using the recently developed causal effect statistical tools. This topic is interesting and important due to the importance of the ISM in climate system and the urgent need of quantification of causal effects. However, there're still some major issues in this manuscript that prevent the acceptance for publication. The biggest problem is that the authors sometimes directly follow previous works' definition and analysis procedure without their own physical thinking.

Major comments:

1. First if focusing on sub-seasonal variability, time filter is first needed to be applied to every dataset: either conducting band-pass filter like the MJO index or using similar time filter process in DW2007. Hypotheses in DW2005 and DW2007 are based on interannual and intraseasonal variability separately, whereas the results obtained in this study is based on weekly data that mix both sub-seasonal and interannual signals, which is very confusing. For example, the pattern in Fig.2d strongly resembles that in DW2005, which is probably due to the dominance of the interannual signal in the unfiltered time series. If the results with filtered data are largely different from that with unfiltered data, the authors' original idea of validating the hypothesis of interannual ISM-CGT relationships (from DW2005) may not be appropriate for this sub-seasonal study.

2. The definition of CGTI is confusing (lines 221). The CGT pattern (or index) defined in DW2005 is on interannual timescale, which is different from the intraseasonal Eurasian wave train pattern presented in DW2007. This study used the exact same region with the interannual CGTI used in DW2005, but the reason for why this region can also be used to define sub-seasonal CGT is not clarified. Again, maybe after filtering, the Z200 pattern does not resemble the CGT pattern and thus the index may not be called as "CGTI".

3. For the EOFs. First, how much of the total variance is explained by each EOF mode, and a test is needed to check whether the EOF modes are well separated from each other (refer to North et al. 1982 and DW2007). Then, comparing Fig. 2c and 2a/b, the Z200 pattern correlated with MT rainfall does not resemble well with the leading two EOF modes (although pattern correlation coefficient is statistically significant, the locations of centers are quite different), indicating this Z200 pattern is not the major mode of global Z200 variability. A suggestion is just focusing on the Eurasian sector (not necessary to link with circumglobal pattern) to conduct EOF, to find key region to define index, and to get wave train pattern.

4. The "PCMCI algorithm and numerical example" section in SI should be added to the Methods in the main text (or replace the original long descriptions). Using equations is much clearer than the original wordy section 2.2 and 2.3. The definition and calculation of the path coefficient (or beta coefficient – unify the term please) should also be added to the Methods section.

5. In Figs.3,5,6,7, what's the meaning of the magnitude of the autocorrelation coefficients? Why can the strength of causal links be expressed by autocorrelation coefficients (lines 257-258)? There's no explanation of these autocorrelation coefficients in the main text.

6. There's a causal arrow from NAO to CGTI in Fig.3, but Fig.4a does not show a NAO-like pattern that is correlated with CGTI. Is NAO a true precursor of the author-defined CGTI?

7. In Fig.5, what's the physical meaning of the arrows from CGTI back to H1 and from L1 back to H1? Statistically, there may have these backward lead-lag correlations, but physically, I don't understand how the change of CGTI/L1 can backward "cause" the change of H1/L1. If these backward arrows are spurious, how can we believe that the other arrows are true causal relations (considering that the authors claimed that their approach can remove the spurious correlations)?

Specific comments:

1. Considering MJO is not a tropical teleconnection, the title of this article may be changed to "Tropical and mid-latitude factors interacting..." (or some other word, not to use "teleconnection").

2. When first using an acronym, a complete spelling should be written before this acronym. E.g., in line 146, "The PC-MCI algorithm" should be changed to "The Peter and Clark – Momentary Conditional Independence (PC-MCI) algorithm".

3. Line 120: Simply explain why choose this box region to define the MT region (e.g.

due to relatively large standard deviation from Fig. 1a). Considering in DW2005 and DW2007, their selected ISM region is to the northwest of this MT region, this again stresses that directly using the hypothesis from these two papers may not be appropriate.

4. Line 420 and Table S2: The explanation of ACE and ACS are not very clear. How large of the magnitude is significant (important)? If ACE and ACS are important expression of the causal effects, Table S2 can be added to the main text. The description of Table S2 writes "actors presented in Fig.8" should be "in Fig.7".

Reference: North, G. R., T. L. Bell, R. F. Cahalan, and F. J. Moeng, 1982: Sampling errors in the estimation of empirical orthogonal functions. Mon. Wea. Rev., 110, 699–706

---

## Author Comment (AC1) · 23 May 2019

Short response to reviewer #1 We thank the anonymous reviewer for his/her comments and suggestions to improve the readability of the manuscript. We will modify the manuscript accordingly to the reviewer's suggestions and we provide here a short response to the main comments together with how we intend to address them in the revised version of the manuscript. 1 - The manuscript is really hard to read because of too many and complex acronyms. I think the authors need to find a way to simplify the reading using nick-names for the tools applied instead of rude acronyms difficult

to remember; Answer - We thank the reviewer for his/her feedback on the readability of the main text. We will work to simplify the acronyms used. 2 - The list of the new findings using these tools should be clearly highlighted in the manuscript, as well as the different weights of the different precursors considered. These capabilities of the tools used are claimed also in the abstract but the results are not clearly extrapolated and summarized in the text; Answer - The new findings of this work are composed of two aspects: first, we prove the hypothesis from D&W2005 from a causal point of view, showing that the expected relationships between the analyzed variables are detected in a causal framework. Second, we quantify the relative importance of the mid-latitude circulation (via CGTI and EOF2), the internal dynamics of the convection cell and Madden and Julia oscillation (MJO) on the Indian summer monsoon (ISM) subseasonal variability (see Figure 7 and table S1 in the SI). We agree that these finding are not as clearly highlighted in the manuscript as it could be. We will rewrite the introduction and discussion to ensure that these main findings are clearly communicated. 3 - Fig 2e,f: how do you explain the propagation in t2m and precipitation? What about winds? Answer - Figures 2e,f do not show the propagation of the signal but only help the reader to visualize the T2m and rainfall anomalies that are linked to high and low circumglobal teleconnection pattern index (CGTI) states. We will clarify this information in the main text. We thank the reviewer for his/her suggestion to show winds anomalies and we will provide the related plot in the revised version of the manuscript. 4 - Figs 3,5,6 and 7: in these type of figures arrows indicate the intensity of the beta coefficients, while the color of the circle the auto-correlations: how are these information combined in interpreting the results? Also what is the real meaning of the intensity of the beta coefficient. It seems in most of the case quite small, thus indicating a very small relationship (?), and it is large only in the case of linking W1 to MT rain (Fig 7) and in MJO2 linking W1 (Fig 7). How are these measures able to weight for the different factors influencing MT rainfall? Answer - We thank the reviewer for pointing out that the definition of beta coefficient is hard to find in the text. Moreover, both terms "beta coefficient" and "path coefficient" refer to the same variable, creating additional confusion. In the revised version of the manuscript, we will move the definition of path coefficient (currently found in lines 266-269) to the method section, and stick to that wording throughout. For clarity, a path coefficient of 0.5 means that a change in the causal parent (e.g. W1) of 1 standard deviation corresponds to a change in 0.5 standard deviation in the response variable (e.g. the MT rainfall). It is correct that the path coefficient between W1 and MJO and the MT rainfall are largest with values of $\sim$0.5 but the path coefficients of the other links are of the same order of magnitude ($\sim$0.2-3) and thus cannot be neglected.

---

## Author Comment (AC2) · 24 May 2019

Response to reviewer #2 We thank the anonymous reviewer for his/her comments and suggestions to improve the readability of the manuscript and the quality of the presented analysis. We will modify the manuscript accordingly to the reviewer's minor comments. Here we provide a short response to the main concerns and outline how we will address them in the revised version of the manuscript. 1. First if focusing on sub-seasonal variability, time filter is first needed to be applied to every dataset: either conducting band-pass filter like the MJO index or using similar time filter process

in DW2007. Hypotheses in DW2005 and DW2007 are based on interannual and intraseasonal variability separately, whereas the results obtained in this study is based on weekly data that mix both sub-seasonal and interannual signals, which is very confusing. For example, the pattern in Fig.2d strongly resembles that in DW2005, which is probably due to the dominance of the interannual signal in the unfiltered time series. If the results with filtered data are largely different from that with unfiltered data, the authors' original idea of validating the hypothesis of interannual ISM-CGT relationships (from DW2005) may not be appropriate for this sub-seasonal study. Answer - We agree that first removing the interannual variability from the timeseries makes a cleaner analyses. We will re-do our full analyses by removing the interannual variability in a pre-processing step as suggested by the reviewer. We will subtract the mean of each season from the weekly anomalies such that the inter-annual variability is removed. 2. The definition of CGTI is confusing (lines 221). The CGT pattern (or index) defined in DW2005 is on interannual timescale, which is different from the intraseasonal Eurasian wave train pattern presented in DW2007. This study used the exact same region with the interannual CGTI used in DW2005, but the reason for why this region can also be used to define sub-seasonal CGT is not clarified. Again, maybe after filtering, the Z200 pattern does not resemble the CGT pattern and thus the index may not be called as "CGTI". Answer - Depending on the outcome of the new analysis (see our response to comment 1), we will decide whether it is appropriate to refer to CGTI as defined in D&W2005 or not. 3. For the EOFs. First, how much of the total variance is explained by each EOF mode, and a test is needed to check whether the EOF modes are well separated from each other (refer to North et al. 1982 and DW2007). Then, comparing Fig. 2c and 2a/b, the Z200 pattern correlated with MT rainfall does not resemble well with the leading two EOF modes (although pattern correlation coefficient is statistically significant, the locations of centers are quite different), indicating this Z200 pattern is not the major mode of global Z200 variability. A suggestion is just focusing on the Eurasian sector (not necessary to link with circumglobal pattern) to conduct EOF, to find key region to define index, and to get wave train pattern. Answer – We

agree with the reviewer on including the information on the total variance explained from each EOF. We have already included the EOF analysis based on the Eurasian sector only (see Figure S2 in the SI) and we show that the two EOFs are very similar to the hemispheric ones shown in the main text (with the only difference that the EOF2 pattern in the hemispheric analyses becomes the EOF1 pattern for the Eurasian sector). We comment these findings in the main manuscript in lines 233-245. Regarding the reviewer's comment that the EOF1 shown in Figure 2a does not resemble the CGT pattern presented in Figure 2d, we would like to highlight that DW2005 also find no resembles between the first EOF and the CGT pattern. Our results are thus consistent with previous findings. The resemblance is however expected for EOF2 (Figure 2b) and the CGT pattern (Figure 2d), which DW2005 interpreted as the second leading mode of the Z200 field in summer. Also, the EOF2 pattern is characterized by 5 positive centers of action that in general correspond to those in the CGT pattern. Still, these findings might be slightly affected by the new pre-processing step (see response to comment 1) and we will thus carefully analyze and communicate this in the revised version. 4. The "PCMCI algorithm and numerical example" section in SI should be added to the Methods in the main text (or replace the original long descriptions). Using equations is much clearer than the original wordy section 2.2 and 2.3. The definition and calculation of the path coefficient (or beta coefficient – unify the term please) should also be added to the Methods section.

Answer – We will include the numerical example in the main manuscript, as suggested by the reviewer.

5. In Figs.3,5,6,7, what's the meaning of the magnitude of the autocorrelation coefficients? Why can the strength of causal links be expressed by autocorrelation coefficients (lines 257-258)? There's no explanation of these autocorrelation coefficients in the main text.

Answer – we thank the reviewer for pointing out the absence of a proper definition of the meaning of auto-correlation coefficient. This definition is similar to the definition of

path coefficient (lines 266-269), in the sense that with autocorrelation we do not mean the usual definition of autocorrelation but the causal influence of the actor on itself, i.e. the amount of change that is attributable to the time series itself.

6. There's a causal arrow from NAO to CGTI in Fig.3, but Fig.4a does not show a NAO like pattern that is correlated with CGTI. Is NAO a true precursor of the author-defined CGTI?

Answer – Fig.4a shows a reversed-sign, NAO-like pattern: A low geopotential height offshore Portugal (L2) and a high geopotential height over Northern Scotland and Iceland (H1). However this is at 200hPa and thus cannot be directly compared to NAO defined at the surface. We thank the reviewer for pointing this out and we will clarify it in the revised manuscript.

7. In Fig.5, what's the physical meaning of the arrows from CGTI back to H1 and from L1 back to H1? Statistically, there may have these backward lead-lag correlations, but physically, I don't understand how the change of CGTI/L1 can backward "cause" the change of H1/L1. If these backward arrows are spurious, how can we believe that the other arrows are true causal relations (considering that the authors claimed that their approach can remove the spurious correlations)?

Answer – We thanks the reviewer for raising this as this is an important aspect of our methodology. The causal discovery algorithm used here extracts the 'causal effect' as defined in a statistical sense (Runge et al. 2015, 2012). This means that auto-correlation effects, common driver effects and indirect links are filtered out. This facilitates the physical interpretation of the network (as compared to a correlation network) but it does not imply that all links are necessary causal in a physical sense. In particular, this can become an issue when actors capture different aspects of one wave pattern (i.e. ridges and crests). In the manuscript, lines 311-322 we try to describe this related to Figure 5: "...Further, the obtained CEN can be interpreted as a wave train that propagates downstream from the east Atlantic towards the monsoon region: when

H1 increases, the CGTI also increases with 1-week lag, while when L1 deepens, CGTI increases. In both cases, this reflects an eastward propagation of a wave from Western Europe via Russia towards the CGTI region. The connections from the CGTI back to H1 and from L1 back to H1 have opposite signs as compared to the downstream links. Thus deepening L1 strengthens the CGTI downstream, but weakens H1 upstream, which is an expression of the eastward propagating wave. A consistent result is found when both L1 and L2 are included in the CEN, although in this case only the downstream causal links are detected (see SI, Fig. S3). In general, this result further confirms the hypothesis proposed by DW2007 that a wave train coming from the North Atlantic influences MT rainfall at about 2-week lead time." Thus, while the CEN can be interpreted physically by a propagating wavetrain, one cannot speak here of cause-and-effect relationships because it is a wavetrain is one dynamical system. Therefore we introduce the term "theory-guided" causal effect network: considering meaningful physical mechanisms has a crucial role while interpreting CENs that represent statistical causal effects. The outcome thus does not undermine the ability of the causal algorithm to detect statistical causal effects, but it stresses the need to accompany statistical tools with a proper understanding of the involved physical mechanisms. We will further clarify this point in the revised manuscript.

References: Runge, J., J. Heitzig, N. Marwan, and J. Kurths, 2012: Quantifying causal coupling strength: A lag-specific measure for multivariate time series related to transfer entropy. Phys. Rev. E, 86, 061121, https://doi.org/10.1103/PhysRevE.86.061121. Runge, J., V. Petoukhov, J. F. Donges, J. Hlinka, N. Jajcay, M. Vejmelka, D. Hartman, N. Marwan, M. Paluš, J. Kurths, 2015: Identifying causal gateways and mediators in complex spatio-temporal systems. Nat. Commun., 6, 9502, https://doi.org/10.1038/ncomms9502.
* * *